# Malnutrition in Infants Aged under 6 Months Attending Community Health Centres: A Cross Sectional Survey

**DOI:** 10.3390/nu13082489

**Published:** 2021-07-21

**Authors:** Carlos S. Grijalva-Eternod, Emma Beaumont, Ritu Rana, Nahom Abate, Hatty Barthorp, Marie McGrath, Ayenew Negesse, Mubarek Abera, Alemseged Abdissa, Tsinuel Girma, Elizabeth Allen, Marko Kerac, Melkamu Berhane

**Affiliations:** 1London School of Hygiene and Tropical Medicine, Keppel Street, London WC1E 7HT, UK; Emma.Beaumont@lshtm.ac.uk (E.B.); marie@ennonline.net (M.M.); Elizabeth.Allen@lshtm.ac.uk (E.A.); Marko.Kerac@lshtm.ac.uk (M.K.); 2UCL Institute for Global Health, 30 Guilford Street, London WC1N 1EH, UK; 3Indian Institute of Public Health, Gandhinagar 382042, Gujarat, India; rrana@iiphg.org; 4GOAL Global, Carnegie House, Library Road, Dun Laoghaire, A96 C7W7 Dublin, Ireland; hbarthorp@goal.ie; 5GOAL Ethiopia, Yeka Subcity, Woreda 9, Addis Ababa P.O. Box 5504, Ethiopia; nahoma@et.goal.ie; 6Emergency Nutrition Network, 69 High Street, Marlborough House, Kidlington, Oxfordshire OX5 2DN, UK; 7Debre Markos University, Debre Markos P.O. Box 269, Ethiopia; ayenewnegesse@gmail.com; 8Jimma University, Jimma P.O. Box 378, Ethiopia; abmubarek@gmail.com (M.A.); alemseged.abdissa@gmail.com (A.A.); tsinuel@yahoo.com (T.G.); melkamuarefayine@gmail.com (M.B.); 9Armauer Hansen Research Institute, Addis Ababa P.O. Box 1005, Ethiopia; 10Harvard Chan School of Public Health, Addis Ababa P.O. Box 1242/5646, Ethiopia

**Keywords:** anthropometric deficit, infants under 6 months, malnutrition, MAMI, Ethiopia, weight-for-age, the Composite Index of Anthropometric Failure

## Abstract

A poor understanding of malnutrition burden is a common reason for not prioritizing the care of small and nutritionally at-risk infants aged under-six months (infants u6m). We aimed to estimate the anthropometric deficit prevalence in infants u6m attending health centres, using the Composite Index of Anthropometric Failure (CIAF), and to assess the overlap of different individual indicators. We undertook a two-week survey of all infants u6m visiting 18 health centres in two zones of the Oromia region, Ethiopia. We measured weight, length, and MUAC (mid-upper arm circumference) and calculated weight-for-length (WLZ), length-for-age (LAZ), and weight-for-age z-scores (WAZ). Overall, 21.7% (95% CI: 19.2; 24.3) of infants u6m presented CIAF, and of these, 10.7% (95% CI: 8.93; 12.7) had multiple anthropometric deficits. Low MUAC overlapped with 47.5% (95% CI: 38.0; 57.3), 43.8% (95% CI: 34.9; 53.1), and 42.6% (95% CI: 36.3; 49.2) of the stunted, wasted, and CIAF prevalence, respectively. Underweight overlapped with 63.4% (95% CI: 53.6; 72.2), 52.7% (95% CI: 43.4; 61.7), and 59.6% (95% CI: 53.1; 65.9) of the stunted, wasted, and CIAF prevalence, respectively. Anthropometric deficits, single and multiple, are prevalent in infants attending health centres. WAZ overlaps more with other forms of anthropometric deficits than MUAC.

## 1. Introduction

Globally, millions of infants aged under six months of age (henceforth infants u6m) are small and nutritionally at-risk, with different forms of anthropometric deficits (e.g., wasted, underweight, or stunted) or low birthweight (LBW) [1]. These deficits are associated with an increased risk of mortality, morbidity, subsequent malnutrition, and impaired development [2]. For instance, an estimated 8.5 million infants u6m are wasted, of whom 3.8 million are severely wasted [3]. In addition, an estimated 20.5 million livebirths have a LBW [4]. These first six months of life are recognised as a crucial developmental period, often with unique nutritional challenges, but with life-long adverse consequences if malnutrition develops [5,6].

Over the last 20 years, community-based management of acute malnutrition (CMAM) has positively transformed the care of malnourished children aged 6–59 months (henceforth children) by relocating the focus of care from inpatient to outpatient settings [7]. This relocation expanded care accessibility and promoted early identification of at-risk children, in turn improving programme coverage and impact [5,8]. Despite these successes, malnourished infants u6m have been left behind [5]. In many settings, they are neither identified nor supported to survive and thrive as per the Every Woman Every Child’s global strategy [5,9]. This is despite evidence of their markedly greater risk of mortality compared with malnourished children [10].

The 2013 update to the WHO guideline for the management of severe malnutrition included for the first time a section on infants u6m, which incorporated recommendations for outpatient care for clinically stable malnourished infants u6m, rather than inpatient care for all, irrespective of clinical status, as was conventionally practiced [11]. However, most low- and middle-income countries (LMICs), have not adopted these updated recommendations into national protocols. Reasons for this include: (i) a poor understanding of the malnutrition burden in infants u6m [5]; (ii) problematic admission criteria based on weight-for-length z-scores (WLZ) [11] that have low reliability [12] and are challenging to obtain in infants u6m, as measuring their length is difficult (more than in children), and WLZ cannot be calculated for lengths < 45 cm [13]; (iii) infants u6m treatment needs are complex [5]; and (iv) the low quality evidence on the best strategies to meet those needs [11].

To address the first two reasons, weight-for-age z-score (WAZ) and the mid-upper arm circumference (MUAC) have been suggested as better alternatives for identifying small and nutritionally at-risk infants u6m, but there is sparse evidence on their prognostic value in this age group. In addition, optimal cut-off values still need to be determined [5,14,15,16]. The growing appreciation of the frequent co-existence of different types of anthropometric deficit further complicates our understanding on the best ways to define malnutrition programme admission criteria [17]. Towards this, the Composite Index of Anthropometric Failure (CIAF) is an aggregated indicator for malnutrition combining the conventional indicators wasted, stunted, and underweight [18,19], but to our knowledge it has not been used to characterise malnutrition in infants u6m.

As in many other LMICs, infant malnutrition is a public health concern in Ethiopia [20]. Current Ministry of Health Guidelines recommend inpatient-only treatment for infants u6m, following identification using a WLZ < −3 [21]. To inform future programme planning and research (including sample size and logistics consideration for a randomised controlled trial (RCT) we are planning in Ethiopia), we aimed to generate caseload-evidence relevant to infants u6m malnutrition. The primary objectives of this study were to assess the prevalence of different indicators of anthropometric deficit in infants u6m attending health centres (this is where small and nutritionally at-risk infants u6m would be screened and recruited into future programmes) and to describe the patterns of overlap between different forms of anthropometric deficit in infants u6m. Our secondary objective was to assess the infant feeding practices of the surveyed infants u6m.

## 2. Materials and Methods

### 2.1. Study Site

The study sites were in Deder woreda, East Hararge zone and in Jimma zone, Ethiopia. Though geographically separate, these are both located in the Oromia region. They were chosen because they are the sites of our future RCT. The study was implemented in 18 health centres, ten in Jimma zone and eight in Deder woreda.

Jimma zone is one of the most populous areas of the Oromia Regional State, with a population of over 3 million people. Deder woreda has a population of some 315,000 people. Both sites have a high burden of malnutrition. Their main livelihood in the area is agriculture, petty trade of cash crops such as khat and coffee, fattening of oxen, and local casual labour. Jimma zone and Deder woreda have 124 and 8 health centres, respectively, each serving an average population of between 15,000 to 30,000.

### 2.2. Study Design and Population

We undertook a health centre-based cross-sectional survey, surveying all infants u6m who attended the selected health centres for any reason over an average period of two weeks in each centre. Reasons for attendance included: being born at the health centre; immunization clinics; growth monitoring clinics; under-5 clinics (where children present with a variety of acute illnesses). We collected data from 1060 infants u6m between 12 October 2020 and 29 January 2021.

### 2.3. Sample Size Estimation

We lacked prior information on how many infants u6m attending the health centres would have anthropometric deficits. Consequently, to estimate a sufficiently robust sample, we assumed a 50% prevalence of anthropometric deficit in infants u6m, and a 3% precision [22]. Using these assumptions, we estimated that we needed a sample of 1067 infants u6m, an average of 60 infants u6m per health centre. To plan for field logistics, we assumed that each health centre would have an average attendance of 30 infants u6m per week and set the average duration of data collection for each health centre to a two-week period.

### 2.4. Health Centre Selection

In Deder woreda, we included all eight available health centres. In Jimma zone, we selected ten out of the 124 available health centres as follows: First, we undertook a register review in all 124 health centres to collect eligibility information on ease of access and patient load. We then excluded 60 health centres from which we were unable to gather complete eligibility data. We further excluded seven health centres that were difficult to access. Lastly, we ranked the remaining 57 health centres according to patient load and randomly selected ten centres from the top 50%.

### 2.5. Training of Data Collectors

We undertook training for the teams of enumerators and supervisors, one team in Jimma zone and one in Deder woreda, to ensure consistency and a high quality of data collection. Our training included learning how to obtain anthropometric measurements (e.g., weight, length, MUAC), assess infant feeding practices, obtain economic and demographic data, use digital data gathering devices (DDGs), as well as to obtain informed consent and clinical history data. Our training also included the piloting of data collection, prior to initiating the actual survey data collection, to ensure the collection of high-quality data and to identify and correct any sources of data collection errors. During the pilot, we also assessed field challenges of the survey tool for final editing.

### 2.6. Data and Measurements

All data were collected using an electronic questionnaire designed using the REDCap (Research Electronic Data Capture) project system (https://redcap.am.lshtm.ac.uk/redcap/ accessed on 14 May 2021).

At the household level, we obtained information about the sex and formal education of the household head, household size, number of dependent children aged < 18 years. From mothers or primary caregivers, we obtained information on age, formal education, and religion.

From all infants u6m, we obtained sex and date of birth data. We asked mothers/caregivers to recall their infants’ age in weeks. We asked whether the infant was born singleton or was a twin or a triplet; the infant’s birth order; how many siblings aged < 18 years they have, and whether any of them had died recently. We collected data on infants’ feeding practices in the past 24 h. We asked about current and past breastfeeding and whether they received any liquids, i.e., water, milk, juice, broth, runny porridge, yoghurt, or other liquids apart from those mentioned. We asked whether they were fed using a bottle and whether they received any solid, semi-solid, or soft foods [23].

We measured weight with the infant undressed using a digital weight scale (Seca 354) to the nearest 5 g if weighing < 10 kg, or to the nearest 10 g if weighing ≥ 10 kg. We measured length using a UNICEF length/height board (infant/child/adult) to the nearest completed 0.1 cm. We measured knee–heel length, from the heel to the upper part of the distal femur (knee), using a digital Vernier calliper (Sealey AK9623EV) to the nearest completed 0.1 cm. We measured MUAC using UNICEF MUAC tapes to the nearest completed 0.1 cm. We measured head circumference using standard anthropometric tapes to the nearest completed 0.1 cm. We obtained subscapular and triceps skinfolds using a Harpenden skinfold calliper to the nearest completed 0.2 mm.

We collected all anthropometric measurements in pairs, as per WHO Child Growth Standards protocols [24]. In brief, for each anthropometric measurement, two enumerators independently took one measurement. If the difference between these two measurements exceeded the pre-set maximum allowed difference (see Appendix B Table A1), each enumerator took a second measurement, and we excluded the first pair of measurements. If the difference for their second pair of measurements also exceeded the allowed difference, they each took a third measurement, and we excluded the first two pairs of measurements. If the difference in their third pair of measurements exceeded the allowed difference, no further measurements were taken, and we excluded all three pairs of measurements.

### 2.7. Data Handling and Analysis

We handled and analysed data using Stata (StataCorp. 2019. Stata Statistical Software: Release 16. College Station, TX, USA: StataCorp LLC).

We estimated infant’s age in weeks as the difference between the survey’s date and the infant’s date of birth divided by seven. For infants whose date of birth was not available, we used maternal recalled age. We grouped infants by age, in 5-week brackets.

We identified infants as being exclusively breastfed if their mothers or primary caregivers reported in their 24 h recall to having breastfed the infants but answered no to all questions regarding any liquid intake, bottle feeding or feeding the infant with solid, semi-solid, or soft foods.

We calculated the mean value for each of the anthropometric measurements using the pair of measurements that did not exceed the allowed difference. We used weight, length, and age data to estimate the anthropometric indices weight-for-age, length-for-age, and weight-for-length z-scores (WAZ, LAZ, and WLZ, respectively) based on the 2006 WHO Child Growth Standards [13], using the *zanthro* Stata command [25]. We marked anthropometric indices as outliers using the 2006 WHO recommendations [26], i.e., if they were >5 or <−6, >6 or <−6, and >5 or <−5 for WAZ, LAZ, and WLZ values, respectively.

We defined underweight, stunted, and wasted as WAZ, LAZ, and WLZ < −2, respectively. We used CIAF to assess overall malnutrition prevalence in infants u6m [18]. We defined CIAF as all infants u6m that were either underweight, stunted, or wasted and we generated the following subcategories: wasted only; wasted and underweight; wasted, stunted and underweight; stunted and underweight; stunted only; and underweight only [18]. We defined the Composite Index of Severe Anthropometric Failure (CISAF) as all infants u6m that were severely underweight, stunted, or wasted, as defined by a WAZ, LAZ, or WLZ < −3, respectively [27].

To explore the overlap between MUAC and CIAF in infants, we explored the use of different thresholds to define low MUAC: MUAC < 11.5 cm; MUAC < 11.0 cm if aged < 6 weeks and <11.5 thereafter; MUAC < 11.0 cm if aged < 7 weeks and <11.5 thereafter; MUAC < 11.0 cm if aged < 13 weeks and <11.5 thereafter; MUAC < 11.0 cm if aged < 17 weeks and <11.5 thereafter; MUAC < 11.0 cm; and MUAC < 10.5 cm. These thresholds were chosen to match those in past/present use in older children; and the age thresholds to match timings of immunization clinic visits when future programmes would use MUAC for the identification of at-risk infants u6m.

For the analysis, we excluded infants u6m for which we could not estimate all anthropometric indices as they had weight or length missing, or their lengths were <45 cm, or if one or more of their anthropometric indices were marked as outliers. In addition, we excluded infants from the assessment of overlap between different anthropometric indicators if they presented with oedema.

We estimated means or proportions, along with the 95% confidence intervals (95% CI), for all variables. We compared basic characteristics between Jimma and Deder using *t*-test and z-test with the *lincom* Stata command. To determine the prevalence of different anthropometric deficits in infants, we estimated summary statistics for all anthropometric variables using the *svy* Stata commands that account for the survey design.

## 3. Results

### 3.1. Participants Flow

Figure 1 shows the survey’s participants flow. Of the infants u6m whose mothers or primary caregivers agreed to participate, a total of 2.17% (95% CI: 1.45; 3.25) had either missing weight or length data, their WLZ could not be estimated because their lengths were <45 cm, or had any of the anthropometric indices WLZ, WAZ, or LAZ identified as outliers.

### 3.2. Sample Characteristics

Key household and caregiver characteristics of 1037 surveyed infants u6m with complete anthropometric information are presented in Table 1. Overall, most households were male-headed and comprised of an average of five members, three of whom were children or adolescents aged <18 years. The infants’ primary caregiver was most often the mother, who was on average in her mid-twenties. Levels of formal education were low: 34% of household heads and 41% of mothers/primary caregivers had no formal education. The infants’ male:female sex ratio was 1.24:1. They were on average 13 weeks of age, and most were singleton births. Mean WAZ, LAZ, and WLZ were all below the 2016 WHO Child Growth Standards median. We found seven infants u6m who presented with oedema: these were excluded from subsequent anthropometric deficits assessment.

We found differences in some of the above-mentioned characteristics between the Jimma and Deder sites: these are presented in Appendix A. The main difference was that infants u6m in Deder presented with significantly worse nutritional status, as evidenced by greater underweight, wasted, and stunted prevalences.

### 3.3. Infant Feeding Practices

A summary of key infant feeding practices is also presented in Table 1. Reported breastfeeding was almost universal with over 98% of mothers reporting that infants u6m had ever been breastfed, 96% receiving breastmilk as the first food after birth, and 95% breastfed in the past 24 h. Almost half of the infants u6m were reported to be exclusively breastfed at the time of assessment and about 15% were bottle-fed. Only 2% were reported to have initiated consumption of any solid, semi-solid, or soft foods.

In Figure 2 we showed the age-smoothed proportion of infants u6m that were exclusively breastfed, were fed different liquids, or were bottle-fed. Median duration of exclusive breastfeeding was 10.6 weeks; and by the age of 16.7 weeks, half of the infants u6m were being given water.

### 3.4. Prevalence of Anthropometric Deficit

Table 2 presents the prevalence of anthropometric deficits in infants u6m, overall and by age categories. We observed that over one in five infants u6m had some form of anthropometric deficit, as indexed by CIAF, and over 4% had a severe anthropometric deficit, as indexed by CISAF. Multiple anthropometric deficits (i.e., wasted and underweight; stunted and underweight; and wasted, stunted, and underweight) affected 10.7% of infants u6m (95% CI: 8.93; 12.7). We did not observe any marked age-dependent variability in the prevalence of infants u6m wasted, stunted, underweight, CIAF, or CISAF.

### 3.5. Prevalence of Low MUAC

Table 3 shows the prevalence of low MUAC using different thresholds. According to the threshold used, we found that overall low MUAC prevalence ranged from 6.71% to 19.1%. We also observed large age-dependent variations: low MUAC was very common in the youngest age groups and decreased sharply in older infants.

Underweight, stunted and wasted was defined as weight-for-age (WAZ), length-for-age (LAZ), and weight-for-length (WLZ) z-scores < −2, respectively. Severe underweight, stunted, and wasted were defined as WAZ, LAZ, and WLZ < −3, respectively. CIAF are all infants u6m that were either underweight, stunted, or wasted. CISAF are all infants u6m that were either severely underweight, stunted or wasted.

### 3.6. Overlap of Low MUAC and Underweight with Different Anthropometric Deficits

In Table 4, we show the proportion of underweight, wasted, stunted, CIAF, and CISAF infants that would be identified by different MUAC thresholds or by underweight if that were used as a sole criterion for the identification of small and nutritionally at-risk infants u6m. Higher MUAC thresholds identify greater proportions of the different definitions of anthropometric deficit. However, they also identified greater proportions of infants u6m that had no anthropometric deficits, as indexed by CIAF. On comparing different MUAC thresholds and underweight for identifying CIAF and CISAF, we observed that underweight identified larger proportions of CIAF and CISAF infants u6m than any MUAC threshold. A visual representation of the overlap, in two different age groups, between the wasted, stunted, and underweight and low MUAC indicators, using a MUAC threshold < 11.0 cm if aged <6 weeks and <11.5 thereafter, is shown in Figure 3.

Appendix A also cross-tabulate MUAC data with CIAF and CIAF categories. The overlap between different MUAC thresholds and CIAF is again observed to be strongly influenced by the infant’s age and how inclusive the MUAC threshold is. For the youngest infants (aged 0–5 weeks), the least inclusive MUAC threshold we examined (<10.5 cm) overlaps with 61% of other forms of CIAF, but also with 40% of infants with no other anthropometric deficits. The most inclusive MUAC threshold (<11.5 cm) overlaps with most (89%) but not all other forms of CIAF, but also with 80% of infants with no other anthropometric deficits. As the age of the infant increases, this proportion of MUAC-identified infants u6m with no CIAF is reduced (to 0% at ages 11–15 weeks), but the overlap with CIAF also decreases. Underweight has a more consistent pattern of overlapping with 39% of all forms of CIAF at age 0–5weeks; with 65% at ages 6–10 weeks; with 52% at ages 11–15 weeks; with 68% at 16–20 weeks and 61% at ages 21–25 weeks.

Low MUAC was defined as MUAC < 11.0 cm if <6 weeks, <11.5 thereafter. Underweight, stunted and wasted was defined as weight-for-age, length-for-age and weight-for-length z-scores < −2, respectively. MUAC: Mid-upper arm circumference.

## 4. Discussion

### 4.1. Summary of Results

To our knowledge, ours is the first study on infants u6m not just in Ethiopia, but in any setting, to estimate the malnutrition burden as indexed by CIAF and its subcategories. The major finding was of a common problem: over 20% of infants u6m attending clinics had some form of anthropometric deficit (CIAF), of which a fifth (4%) were severe (CISAF) and over half (11%) were multiple anthropometric deficits with combinations of wasted, stunted, or underweight.

Estimating malnutrition burden using the simplest index, i.e., MUAC, resulted in a wide prevalence range depending on the infants’ age and the threshold used to define low MUAC. Whilst MUAC-defined prevalence overlapped with CIAF-defined prevalence, the extent of overlap varied markedly and was lowest in the youngest age group where many infants with low MUAC had no other CIAF-defined anthropometric deficit. In contrast to marked age-related differences for MUAC-based estimates, wasted-, stunted-, underweight-, CIAF-, and CISAF-based estimates were similar across the different age categories. There were no consistent upwards or downwards prevalence trends across the age categories. Underweight as the sole case definition criterion overlapped with at least half of the infants u6m with other forms of anthropometric deficit and identified larger proportions of CIAF and CISAF infants than any MUAC threshold.

### 4.2. Burden of Malnutrition and Programme Implications

Even though our study used a health centre-based rather than population-based sample, our observed wasted and underweight prevalence are comparable with recent national estimates from the 2019 Ethiopian Mini DHS. However, in our sample, infants u6m were markedly less stunted than the national prevalence (9.8% versus 17.1%) [20]. Our data on the overlap between different types of anthropometric deficit were also consistent with those from older children where this has been observed, and where there is current focus on the overlap between wasting and stunting, since this can greatly increase mortality risk [28]. In older children, there is also increased focus on underweight as a simple single measure and way of capturing this overlap without having to assess height-based indices [28].

Current Ethiopian guidelines (and most other national guidelines which are also based on the 2013 update to the WHO guideline for management of severe malnutrition) for admitting infants u6m to malnutrition treatment programmes focus only on those who are severely wasted, as indicated by a WLZ < −3 [21]. Our data show that this focus misses many infants u6m with other forms of anthropometric deficit. Widening admission criteria would certainly have programme caseload and capacity implications. Our results showed that the number of admissions would almost double (from 2.4% to 4.3%) if CISAF was to become the criterion for admission, or it would increase by ten-fold if the criteria was any form of anthropometric deficit as denoted by CIAF. Key questions for future research are what impact there would be on programme outcomes, notably mortality and morbidity, whether the caseload expansion would be justified by the benefits, and whether it would be cost-effective. A closely related question is what the benefits of a shift to outpatient-based care for at-risk infants u6m are. This alone would markedly improve programme capacity over current inpatient-only models of care.

### 4.3. The Utility of Simple Measurements: MUAC and WAZ

A small but growing number of studies have explored the utility of different anthropometric criteria to identify malnutrition in infants u6m [29,30,31,32,33,34,35,36,37]. The best of these look at the association between anthropometric measures and subsequent mortality. This conceptual approach recognises that what matters is clinical outcome rather than body size, that there is no one anthropometric measure considered the ‘gold standard’ for identifying malnutrition, and that all anthropometric measures are only proxy measures of nutritional status, each with advantages and disadvantages and differing in sensitivity, specificity, and positive and negative predictive value when identifying infants u6m at-risk of adverse outcomes, notably mortality and morbidity [17]. Despite differing designs and contexts, most of these studies on infants u6m agree that WLZ is poor at identifying high-risk infants and that WAZ or MUAC are better and more practical. WAZ is already widely measured in growth monitoring programmes so would be particularly easy to adopt. Whilst our cross-sectional study design could not assess the prognostic value of these two indices in our population, we have shown the potential caseload implications.

The extent to which low MUAC, especially in the first six weeks of life, overlaps with LBW is a major unknown. This overlap is plausible and matters because LBW is a well-established risk factor for both short- and long-term mortality and morbidity [38], even when other anthropometric indicators are within normal range [30]. If a large overlap between low MUAC and LBW in the first weeks of life is present, it would strongly support the use of low MUAC to identify infants u6m for enrolment to treatment/support programmes at this age, even when the overlap with other anthropometric deficits is poor. A recent study of newborns in Ethiopia found a MUAC ≤ 9.8 cm, measured within 24 h of life, to be a useful diagnostic tool for LBW [39]. Conversely, if there is a poor overlap between low MUAC and LBW, then there might be less value in using low MUAC, as these infants might have smaller arms because they are young and hence small.

Either way, future research is needed to assess the independent role of MUAC in identifying infants u6m at high risk of mortality/morbidity. MUAC-for-age tables are available and might be used to improve the predictive value of low MUAC, but this might work against the practical and programmatic advantage of MUAC, namely simplicity and speed of assessment.

Much current research on anthropometric deficits as a risk factor has focused on how well it identifies individuals at risk of short-term morbidity or mortality [40]. However, less attention has been given to assess how well it predicts a longer-term risk of impaired physical or cognitive development. Evidence from a recent systematic review shows that adverse nutrition in infancy and childhood is associated with long-term adult non-communicable disease [6]. Future research should explore how both MUAC and the various subcategories of CIAF predict these longer-term risks and possible underlying mechanisms and pathways. The role of body composition and body proportions, both indicators of development, could help explain what lies behind these simple anthropometric measurements. Past research has shown that MUAC and WLZ have different associations with body composition, where WLZ appears to have similar associations with lean and fat mass, whilst it also has a small but at times negligible association with length. MUAC, on the other hand, appears to have a stronger association with fat mass than with lean mass, but also has a strong association with length, i.e., each index MUAC and WLZ appears to select infants u6m for different phenotypes [41].

### 4.4. Implications for Research and Programming

Our data have immediate implications for programming and research using a new “management of small and nutritionally at-risk mothers and infants (MAMI)” Care Pathway, just launched in May 2021 [1]. Building on past experiences [29], the MAMI Care Pathway emphasises the potential of MUAC as an independent programme enrolment criterion. Our results highlight that more data are needed to understand the prognostic utility of MUAC, especially in the youngest infants u6m where overlap with other forms of anthropometric deficits are the least. Balancing the need for simplicity and recognising the rapid increase in MUAC, especially in the first month of life, our own planned research in Ethiopia will now include both single indicators, a WAZ < −2 and a MUAC < 11.0 cm if aged < 6 weeks and <11.5 thereafter, whereby of the total sample of included infants u6m, about 26% of them will be identified by both indicators.

### 4.5. Strengths and Limitations

We recognise the limitations of our work. Most importantly, our data from health centres might not be representative of the wider infants u6m population. However, these data remain important since recruitment into future programmes is likely to occur at the health centre level and hence our data matter for the planning of such programmes. Existing programmes such as immunization have very high coverage and thus our strategy of screening at such visits would be both efficient and would identify many high-risk infants u6m early on. Related to this, our health centre selection was not randomised and did not seek to be representative of all the health centres in the Oromia region. We selected based on ease of access and on the number of infants u6m who receive care, informed by registered data. Whilst this limits our ability to generalise the absolute prevalence results to the wider region/country, there is no reason to believe that the overlaps between indicators and the age-related variations should be very different elsewhere. Nonetheless, this should be examined in future work.

As already noted, missing birthweight data represents an important limitation. Whilst we tried to collect birthweight data, this was unavailable for most infants u6m. As a result, we were unable to disentangle how much of the anthropometric deficit we observed could be explained by poor foetal growth, and which of the different indicators of anthropometric deficits are best at identifying small infants that were LBW.

In this analysis, we also did not analyse the reasons underlying or associated with anthropometric deficit. This should be the focus for future work. A particularly important future question is how much anthropometric deficit growth failure is reversible with nutritional and other interventions and what are the clinical and functional benefits (e.g., on mortality, morbidity, and child development) of these interventions. These questions we seek to address in our own planned RCT, and we hope this paper will inspire others to do similar work in other settings.

Finally, we did not stratify our analysis by the infant’s sex and, as such, we are not able to understand whether our findings differ by sex. Data from children have shown that boys are often more underweight, wasted, and stunted, whereas there is a sex bias towards girls when identifying malnutrition using MUAC [42,43]. Nonetheless, none of these limitations are likely to invalidate the findings of our study, rather it forms the basis of future work.

Our study also has strengths. In our study, we sampled infants u6m attending health centres for any reason. Most studies evaluating malnutrition burden in infants u6m focused on community representative samples [29,32,33,34], on infants hospitalised and receiving in-patient care [31,35,36,37], or they have followed-up a birth cohort [30]. Our study was a facility-based survey collecting data on infants attending health centres for any reason, which provides better understanding of the potential for care provision to the population covered by these health centres. Future enrolment into programmes for small and nutritionally at-risk infants u6m is likely to occur at health centres, so it is vital to know numbers at these facilities.

We answered a recent call to use aggregate measures to quantify malnutrition and the burden of malnutrition as any manifestation of anthropometric deficit as indexed by CIAF [17]. Published research on malnutrition in infants u6m has focused strongly on severe wasting as denoted by a WLZ < −3 [30,31,34,37], but our data add to growing calls for more inclusive case definitions for identifying infants at risk of malnutrition, mortality, or impaired development. As mentioned above, we are currently developing an intervention trial that will test the benefit of a more inclusive approach and we hope that our data will encourage others to do likewise. The analysis in the present study provides the first step towards quantifying the burden of CIAF and of MUAC in infants u6m and the degree of convergence between indicators and across age subgroups.

## 5. Conclusions

Single and multiple anthropometric deficits are prevalent in Ethiopian infants u6m attending health centres. To identify any form of anthropometric deficit, as classified by wasted, stunted, underweight, CIAF, or CISAF, WAZ appears to perform better than MUAC, whilst both are good at identifying infants u6m with multiple anthropometric deficits. Further research is needed to understand which criteria or combination of criteria would be best for future programmes managing small and nutritionally at-risk infants u6m and to understand the associated functional and clinical outcomes, notably short term-risks of mortality and morbidity.

## Figures and Tables

**Figure 1 nutrients-13-02489-f001:**
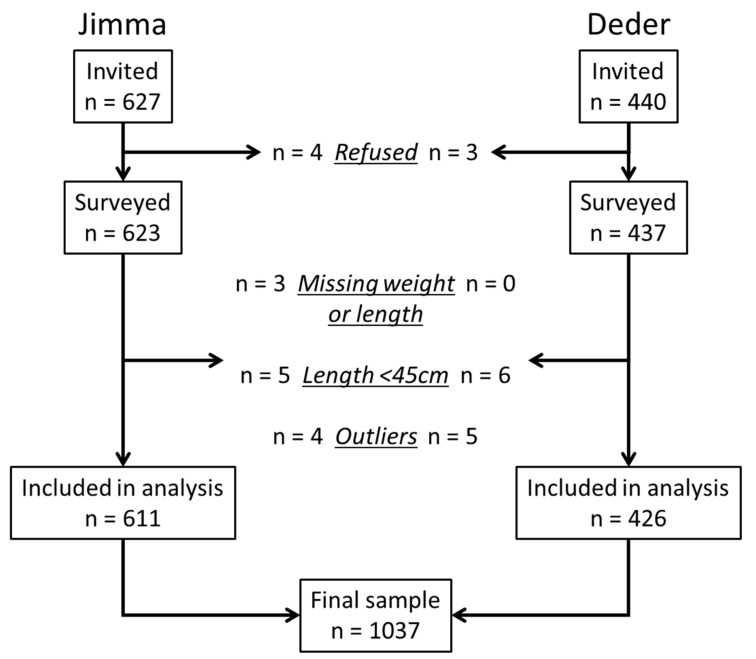
Study participants’ flow chart.

**Figure 2 nutrients-13-02489-f002:**
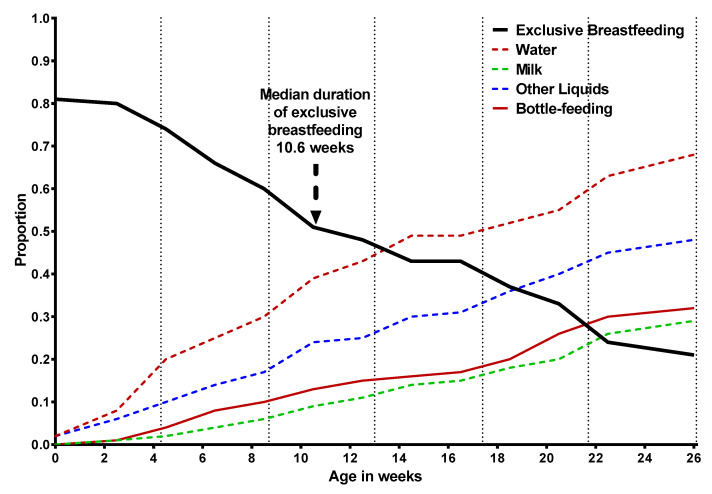
Age-smoothed proportion of exclusively breastfed infants aged under six months. The figure also presents age-smoothed proportions of bottle feeding and the consumption of water, milk and other liquids (other than water, milk, juice, broth, runny porridge, or yoghurt).

**Figure 3 nutrients-13-02489-f003:**
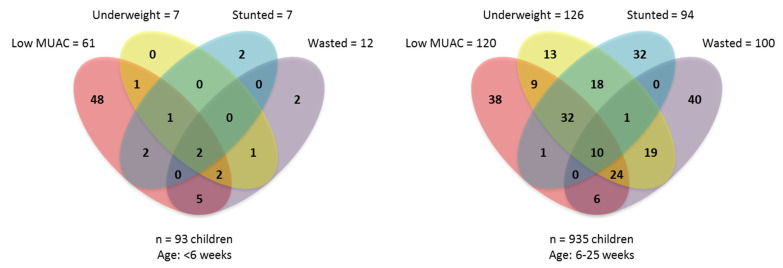
A Venn-diagram showing the overlap of different malnutrition indicators in two age groups.

**Table 1 nutrients-13-02489-t001:** Characteristics of the 1037 surveyed infants under six months of age.

	Mean Or %	95% CI
**Household Characteristics**		
Household members	5.4	5.3; 5.6
Household children aged <18 years	3.2	3.1; 3.3
Household head is male (%)	98.9	98.1; 99.4
Household head formal education		
No education (%)	33.9	31.0; 36.9
Grade 1–8 (%)	45.5	42.4; 48.6
Grade 9–12 (%)	14.0	11.9; 16.3
Technical and Vocational Education (%)	0.60	0.30; 1.40
College/University (%)	6.10	4.70; 7.70
**Mother/Caregiver Characteristics**		
Mother as main caregiver (%)	99.2	98.5; 99.6
Grandparent help in infant care (%)	35.8	32.9; 38.7
Mother/Caregiver age (years)	25.9	25.6; 26.2
Mother/Caregiver is married (%)	98.8	98.0; 99.3
Mother/Caregiver age at marriage (years)	17.5	17.3; 17.6
Time married (years)	8.45	8.10; 8.79
Mother/Caregiver formal education		
No education (%)	40.9	37.9; 43.9
Grade 1–8 (%)	45.8	42.8; 48.9
Grade 9–12 (%)	9.16	7.55; 11.1
Technical and Vocational Education (%)	1.25	0.73; 2.15
College/University (%)	2.89	2.03; 4.11
Mother/Caregiver religion		
Muslim (%)	94.2	92.6; 95.5
Orthodox Christian (%)	4.60	3.50; 6.10
Protestant (%)	1.10	0.60; 1.90
Refused to answer (%)	0.10	0.00; 0.70
**Infant Characteristics**		
Infant’s age (weeks)	13.4	13.0; 13.8
Infant is male (%)	55.4	52.3; 58.4
Infant is singleton (%)	98.6	97.6; 99.1
Siblings aged <18 years	2.2	2.0; 2.3
Infant’s birth order		
1st (%)	25.0	22.4; 27.7
2nd (%)	20.6	18.3; 23.2
3rd (%)	15.0	13.0; 17.4
4th (%)	12.3	10.4; 14.4
5th (%)	11.3	9.49; 13.4
6th+ (%)	15.8	13.7; 18.2
Recent death of sibling (%)	14.9	12.8; 17.2
**Infant feeding**		
Infant ever breastfed (%)	98.9	98.1; 99.4
Infant received breastmilk as first food (%)	96.1	94.7; 97.1
Infant was breastfed in the past 24 h (%)	94.1	92.5; 95.4
Breastfeeding frequency in past 24 h (times)	10.4	10.3; 10.6
Infant exclusively breastfed (%)	48.9	45.9; 51.9
Infant was bottle-fed (%)	15.2	13.2; 17.6
Infant fed any solid, semi-solid or soft foods (%)	2.03	1.32; 3.09
**Infant anthropometry**		
Bilateral pitting oedema (%)	0.68	0.32; 1.41
Weight (kg)	5.62	5.55; 5.70
Length (cm)	59.6	59.4; 59.9
Mid-upper arm circumference (cm)	12.4	12.4; 12.5
Triceps skinfold (mm)	7.7	7.6; 7.8
Subscapular skinfold (mm)	6.8	6.7; 6.9
Head circumference (cm)	40.4	40.2; 40.5
Knee-to-heel length (mm)	148	147; 149
Weight-for-age z-score	–0.65	–0.72; –0.57
Length-for-age z-score	–0.35	–0.43; –0.26
Weight-for-length z-score	–0.47	–0.55; –0.40

**Table 2 nutrients-13-02489-t002:** Proportion of infants aged under six months with different indicators of anthropometric deficit.

	All		0–5 Weeks		6–10 Weeks		11–15 Weeks		16–20 Weeks		20–25 Weeks	
	*n* = 1030		*n* = 93		*n* = 268		*n* = 285		*n* = 243		*n* = 141	
Anthropometric Indicator	%	95% CI	%	95% CI	%	95% CI	%	95% CI	%	95% CI	%	95% CI
Wasted	10.9	9.11; 12.9	12.9	7.47; 21.4	8.21	5.46; 12.2	10.2	7.16; 14.3	12.8	9.11; 17.6	12.8	8.19; 19.4
Severely wasted	2.43	1.64; 3.57	--	--	--	--	--	--	--	--	--	--
Stunted	9.81	8.13; 11.8	7.53	3.63; 15.0	9.70	6.69; 13.9	9.12	6.28; 13.1	10.7	7.38; 15.3	11.4	7.07; 17.7
Severely stunted	3.47	2.51; 4.78	--	--	--	--	--	--	--	--	--	--
Underweight	12.9	11.0; 15.1	7.53	3.63; 15.0	13.8	10.2; 18.5	10.5	7.45; 14.7	16.5	12.3; 21.7	13.5	8.76; 20.2
Severely underweight	4.56	3.44; 6.02	--	--	--	--	--	--	--	--	--	--
CIAF	21.7	19.2; 24.3	19.4	12.5; 28.7	21.3	16.8; 26.6	20.4	16.1; 25.4	24.3	19.3; 30.1	22.0	15.9; 29.6
No CIAF	78.4	75.7; 80.8	80.7	71.3; 87.5	78.7	73.4; 83.2	79.7	74.6; 83.9	75.7	69.9; 80.7	78.0	70.4; 84.1
*CIAF categories*												
Wasted only	5.15	3.95; 6.68	--	--	--	--	--	--	--	--	--	--
Wasted and Underweight	4.47	3.36; 5.91	--	--	--	--	--	--	--	--	--	--
Wasted, Stunted and Underweight	1.26	0.73; 2.16	--	--	--	--	--	--	--	--	--	--
Stunted and Underweight	4.95	3.78; 6.46	--	--	--	--	--	--	--	--	--	--
Stunted only	3.59	2.61; 4.92	--	--	--	--	--	--	--	--	--	--
Underweight only	2.23	1.49; 3.34	--	--	--	--	--	--	--	--	--	--
CISAF	4.27	3.19; 5.69	1.08	0.15; 7.25	4.85	2.84; 8.18	5.61	3.47; 8.97	5.35	3.13; 9.00	0.71	0.10; 4.87

CIAF: Composite Index of Anthropometric Failure. CISAF: Composite Index of Severe Anthropometric Failure. Underweight, stunted and wasted was defined as weight-for-age (WAZ), length-for-age (LAZ) and weight-for-length (WLZ) z-scores < −2, respectively. Severe underweight, stunted and wasted was defined as WAZ, LAZ and WLZ < −3, respectively. CIAF are all infants u6m that were either underweight, stunted or wasted. CISAF are all infants u6m that were either severely underweight, stunted or wasted.

**Table 3 nutrients-13-02489-t003:** Proportion of infants aged under six months with low MUAC as defined by different thresholds.

	All		0–5 Weeks		6–10 Weeks		11–15 Weeks		16–20 Weeks		20–25 Weeks	
	*n* = 1028		*n* = 93		*n* = 267		*n* = 285		*n* = 243		*n* = 140	
MUAC Indicator	%	95% CI	%	95% CI	%	95% CI	%	95% CI	%	95% CI	%	95% CI
MUAC < 10.5 cm	6.71	5.33; 8.42	44.1	34.4; 54.3	4.12	2.29; 7.29	3.16	1.65; 5.96	2.47	1.11; 5.39	1.43	0.36; 5.54
MUAC < 11.0 cm	12.0	10.1; 14.1	65.6	55.4; 74.5	11.6	8.28; 16.0	5.26	3.20; 8.55	4.53	2.52; 7.99	3.57	1.49; 8.30
MUAC < 11.0 cm if <17 weeks, <11.5 thereafter	12.7	10.8; 14.9	65.6	55.4; 74.5	11.6	8.28; 16.0	5.26	3.20; 8.55	5.76	3.44; 9.50	7.14	3.88; 12.8
MUAC < 11.0 cm if <13 weeks, <11.5 thereafter	13.4	11.5; 15.7	65.6	55.4; 74.5	11.6	8.28; 16.0	7.72	5.13; 11.5	5.76	3.44; 9.50	7.14	3.88; 12.8
MUAC < 11.0 cm if <7 weeks, <11.5 thereafter	16.5	14.4; 18.9	65.6	55.4; 74.5	20.6	16.2; 25.9	10.5	7.45; 14.7	5.76	3.44; 9.50	7.14	3.88; 12.8
MUAC < 11.0 cm if <6 weeks, <11.5 thereafter	17.6	15.4; 20.1	65.6	55.4; 74.5	24.7	19.9; 30.3	10.5	7.45; 14.7	5.76	3.44; 9.50	7.14	3.88; 12.8
MUAC < 11.5 cm	19.1	16.8; 21.6	81.7	72.5; 88.3	24.7	19.9; 30.3	10.5	7.45; 14.7	5.76	3.44; 9.50	7.14	3.88; 12.8

MUAC: Mid-upper arm circumference.

**Table 4 nutrients-13-02489-t004:** Overlap of different anthropometric deficits with different low MUAC thresholds and Underweight.

	Underweight		Stunted		Wasted		CIAF		CISAF		No CIAF	
	*n* = 133		*n* = 101		*n* = 112		*n* = 223		*n* = 44		*n* = 805	
Indicator	%	95% CI	%	95% CI	%	95% CI	%	95% CI	%	95% CI	%	95% CI
MUAC < 10.5cm	24.1	17.6; 32.1	22.8	15.6; 32.0	17.0	11.1; 25.1	17.0	12.7; 22.6	36.4	23.6; 51.4	3.85	2.72; 5.43
MUAC < 11.0 cm	44.4	36.2; 52.9	38.6	29.6; 48.4	27.7	20.2; 36.7	30.5	24.8; 36.9	43.2	29.5; 58.0	6.83	5.28; 8.80
MUAC < 11.0 cm if <17 weeks, <11.5 thereafter	50.4	41.9; 58.8	42.6	33.3; 52.4	33.9	25.8; 43.2	34.1	28.2; 40.6	43.2	29.5; 58.0	6.83	5.28; 8.80
MUAC < 11.0 cm if <13 weeks, <11.5 thereafter	54.1	45.6; 62.4	43.6	34.2; 53.4	36.6	28.2; 45.9	36.3	30.3; 42.9	45.5	31.5; 60.2	7.08	5.50; 9.07
MUAC < 11.0 cm if <7 weeks, <11.5 thereafter	60.9	52.4; 68.8	47.5	38.0; 57.3	43.8	34.9; 53.1	42.6	36.3; 49.2	45.5	31.5; 60.2	9.32	7.49; 11.5
MUAC < 11.0 cm if <6 weeks, < 11.5 thereafter	60.9	52.4; 68.8	47.5	38.0; 57.3	43.8	34.9; 53.1	42.6	36.3; 49.2	45.5	31.5; 60.2	10.7	8.73; 13.0
MUAC < 11.5cm	61.7	53.1; 69.5	47.5	38.0; 57.3	46.4	37.4; 55.7	44.0	37.6; 50.5	45.5	31.5; 60.2	12.2	10.1; 14.6
Underweight	--	--	63.4	53.6; 72.2	52.7	43.4; 61.7	59.6	53.1; 65.9	65.9	50.9; 78.3	--	--

CIAF: Composite Index of Anthropometric Failure. CISAF: Composite Index of Severe Anthropometric Failure. Underweight, stunted and wasted was defined as weight-for-age (WAZ), length-for-age (LAZ) and weight-for-length (WLZ) z-scores < −2, respectively. Severe underweight, stunted and wasted was defined as WAZ, LAZ and WLZ < −3, respectively. CIAF are all infants u6m that were either underweight, stunted or wasted. CISAF are all infants u6m that were either severely underweight, stunted or wasted.

## Data Availability

The data presented in this study are available on request from the corresponding author.

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
