# Peer review of "Malnutrition in Infants Aged under 6 Months Attending Community Health Centres: A Cross Sectional Survey"

_nutrients, 2021, doi:10.3390/nu13082489_

Round 1

Reviewer 1 Report

This is a very interesting and important topic. However, insufficient information has been provided.
Comments 1: Line 114: Where did you get the percentage of 50% prevalence of anthropometric deficit in infants u6m, and a 3% precision?  Any reference?? A more detailed explanation is required.
Comments 2: IRB approval information obtaining consent are missing.
Comments 3: Table1. Demographic information table – no information of house income
Comments 4: Delivery modes, current disease status could be a significant factor for these babies’ growth.  
Comments 5: Line 232: what is indicated lower or high education in this study, please define
Comments 6: It could be interesting if you look at MUAC and other anthropometric measures including Head circumference, Knee-heel length, subscapular skinfold thickness, triceps skinfold thickness which was already measured in your study but did not indicate the manuscript. These are comment anthropometric measures for nutrition as well.

Reviewer 2 Report

The authors present important findings on the prevalence of malnutrition in young infants in a low/middle income country. They highlight the important issue of a lack of overlap between different anthropometric parameters, complicating the definition of malnutrition in young children. It would be interesting to see how the different MUAC and weight for age, length for age and weight for length parameters correlate with low fat parameters, since skin fold measurements were performed as well.

When estimating the prevalences, the centres should be added as a random effect in the analyses. How many of the children had a birth date that was by parental report only ?
